# Combination Assessment of Diffusion-Weighted Imaging and T2-Weighted Imaging Is Acceptable for the Differential Diagnosis of Lung Cancer from Benign Pulmonary Nodules and Masses

**DOI:** 10.3390/cancers13071551

**Published:** 2021-03-28

**Authors:** Katsuo Usuda, Masahito Ishikawa, Shun Iwai, Yoshihito Iijima, Nozomu Motono, Munetaka Matoba, Mariko Doai, Keiya Hirata, Hidetaka Uramoto

**Affiliations:** 1Department of Thoracic Surgery, Kanazawa Medical University, Ishikawa 920-0293, Japan; masa-i@kanazawa-med.ac.jp (M.I.); mhg1214@kanazawa-med.ac.jp (S.I.); y-iijima@kanazawa-med.ac.jp (Y.I.); motono@kanazawa-med.ac.jp (N.M.); hidetaka@kanazawa-med.ac.jp (H.U.); 2Department of Radiology, Kanazawa Medical University, Ishikawa 920-0293, Japan; m-matoba@kanazawa-med.ac.jp (M.M.); doaimari@kanazawa-med.ac.jp (M.D.); 3MRI Center, Kanazawa Medical University Hospital, Ishikawa 920-0293, Japan; keiya@kanazawa-med.ac.jp

**Keywords:** magnetic resonance imaging (MRI), diffusion-weighted magnetic resonance imaging (DWI), T2-weighted imaging (T2WI), lung cancer, pulmonary nodule and mass (PNM), pulmonary abscess, apparent diffusion coefficient (ADC)

## Abstract

**Simple Summary:**

The purpose of this study is to determine whether the combination assessment of DWI and T2WI improves the diagnostic ability for differential diagnosis of lung cancer from benign pulmonary nodules and masses (BPNMs). As using the OCV (1.470 × 10^−3^ mm^2^/s) for ADC, the sensitivity was 83.9% (220/262), the specificity 63.4% (33/52), and the accuracy 80.6% (253/314). As using the OCV (2.45) for T2 CR, the sensitivity was 89.7% (235/262), the specificity 61.5% (32/52), and the accuracy 85.0% (267/314). In 212 PNMs which were judged to be malignant by both DWI and T2WI, 203 PNMs (95.8%) were lung cancers. In 33 PNMs which were judged to be benign by both DWI and T2WI, 23 PNMs (69.7%) were BPNMs. The combined assessment of DWI and T2WI could judge PNMs more precisely and would be acceptable for differential diagnosis of PNMs.

**Abstract:**

The purpose of this study is to determine whether the combination assessment of DWI and T2-weighted imaging (T2WI) improves the diagnostic ability for differential diagnosis of lung cancer from benign pulmonary nodules and masses (BPNMs). The optimal cut-off value (OCV) for differential diagnosis was set at 1.470 × 10^−3^ mm^2^/s for apparent diffusion coefficient (ADC), and at 2.45 for T2 contrast ratio (T2 CR). The ADC (1.24 ± 0.29 × 10^−3^ mm^2^/s) of lung cancer was significantly lower than that (1.69 ± 0.58 × 10^−3^ mm^2^/s) of BPNM. The T2 CR (2.01 ± 0.52) of lung cancer was significantly lower than that (2.74 ± 1.02) of BPNM. As using the OCV for ADC, the sensitivity was 83.9% (220/262), the specificity 63.4% (33/52), and the accuracy 80.6% (253/314). As using the OCV for T2 CR, the sensitivity was 89.7% (235/262), the specificity 61.5% (32/52), and the accuracy 85.0% (267/314). In 212 PNMs which were judged to be malignant by both DWI and T2WI, 203 PNMs (95.8%) were lung cancers. In 33 PNMs which were judged to be benign by both DWI and T2WI, 23 PNMs (69.7%) were BPNMs. The combined assessment of DWI and T2WI could judge PNMs more precisely and would be acceptable for differential diagnosis of PNMs.

## 1. Introduction

Pulmonary nodules and masses (PNMs) are common medical problems for which management can be quite complex. Among the PNMs, lung cancer is the most serious disease related to deaths, and its correct diagnosis is essential for patients. CT is the imaging standard of choice for assessing PNMs in daily practice. 18-fluoro-2-deoxy-glucose positron emission tomography/computed tomography (FDG-PET/CT) is valuable for discriminating lung cancer from benign pulmonary nodules and masses (BPNMs) [1]. FDG-PET/CT has been widely adopted for its imaging modality of PNMs. Its maximum standardized uptake value (SUVmax) shows glucose uptake and expresses how invasive the cancer is. However, FDG-PET/CT is very costly, and its assessment is difficult because there are cases of false-negative results for well-differentiated pulmonary adenocarcinomas [2] and small volumes of metabolically active tumors [3], and false-positive results for inflammatory nodules [4].

In the past, the use of pulmonary magnetic resonance imaging (MRI) has been limited by several weaknesses, including low signal-to-noise ratios due to low proton density in the lungs, loss of signal related to cardiac and respiratory motion, and artifacts related to the magnetic susceptibility differences between air and soft tissue. MRI for lung cancer has been limitedly performed in the cases of mediastinum invasion or chest wall invasion of lung cancer partly due to the report of Webb et al. of the Radiologic Diagnostic Oncology Group in 1991 [5]. Recently MRI technology has advanced dramatically. Diffusion-weighted magnetic resonance imaging (DWI) has been applied both to discriminate PNMs and to evaluate response evaluation of treatment for lung cancer. DWI has been utilized for detecting the free diffusion of water molecules (Brownian motion) with areas of restricted diffusion. Its apparent diffusion coefficient (ADC) value shows a quantitative parameter of the diffusion of water molecules in biological tissues, and the ADCs of malignant tumors are significantly lower than those of normal tissues or benign lesions [6]. All the meta-analyses mentioned that DWI could differentiate malignancy from benignity for PNMs [7,8,9] and might decrease unnecessary biopsy rates and complications of biopsies [10].

In practice, it is sometimes difficult to differentiate a pulmonary abscess from lung cancer by DWI because a pulmonary abscess shows strongly restricted diffusion like lung cancer. T2-weighted imaging (T2WI) is an essential MRI examination. DWI is related to T2WI and influenced by T2-shine through. The usefulness of T2WI was demonstrated, especially in the evaluation of high-intensity fluid materials associated with the organ lesions, such as intratumoral necrosis, cysts, mucus, hemorrhage, or edema [11,12]. Actually, T2WI is reliable for the diagnosis of cystic mediastinal tumors. Lung cancer is likely to show lower intensity compared to BPNMs in T2WI.

Although DWI can be valuable for differential diagnosis of lung cancer from BPNMs, the diagnostic capability may not be perfect. The interpretation of small metastatic nodules, nonsolid adenocarcinoma, some granulomas, and active inflammatory nodules should be approached with caution [13].

The purpose of this study is to determine whether the combination assessment of DWI and T2WI improves the diagnostic ability for discriminating lung cancer from BPNMs.

## 2. Materials and Methods

### 2.1. Eligibility

The study protocol for examining MRI in patients with PNMs was approved by the ethical committee of Kanazawa Medical University (the approval number: No. I302). Written informed consent for MRI was obtained from each patient after discussing the risks and benefits of the examinations. The study was conducted according to the guidelines of the Declaration of Helsinki.

### 2.2. Patients

Patients who had lung cancer or a BPNM in chest X-ray were examined with plain CT first. The primary diagnosis of lung cancer or BPNM was performed by chest CT. In the patients who had primary lung cancers or BPNMs in CT, and had MRI examinations before pathological diagnosis and bacterial diagnosis from May 2009 to April 2020, 314 patients qualified for detailed analysis of MRI with T2WI and DWI (Table 1). Patients included in the study had PNMs with a maximum size of 150 mm or less (range 5–150 mm, mean 31.9 mm) in CT, and which had no definitive calcification. Most of the PNMs were pathologically diagnosed by resection, or through a flexible bronchoscopy. The other remaining PNMs were diagnosed by bacterial culture or roentgenographically follow-up study. The PNMs were diagnosed as benign when the PNMs decreased in size or disappeared upon review of retrospective chest x-ray films or CT. Pure ground-glass-opacity (GGO)-type lung cancers were excluded from this study. None of the patients had received prior treatment. 196 patients were male and 118 were female. Their mean age was 68 years old (range 37 to 85). There were 262 lung cancers and 52 BPNMs. Out of 262 patients with lung cancer, 226 of those patients that were enrolled in this study were also used in another paper [14]. The diagnosis was made pathological in all lung cancers. For 262 lung cancers, there were 183 adenocarcinomas, 60 squamous cell carcinomas, 4 large cell neuroendocrine carcinomas (LCNECs), 3 large cell carcinomas, 3 adenosquamous carcinomas, 2 carcinoids, 6 small cell carcinomas and 1 carcinosarcoma. TNM classification and the lymph node stations of lung cancer were classified according to the new definitions in UICC 8 [15]. There were 2 pathological T1mi (pT1 mi) carcinomas, 70 pT1a carcinomas, 49 pT1b carcinomas, 3 pT1c carcinomas, 78 pT2a carcinomas, 19 pT2b carcinomas, 35 pT3 carcinomas, and 6 pT4 carcinomas. There were 209 pathological N0 (pN0) carcinomas, 33 pN1 carcinomas, and 20 pN2 carcinomas. There were 254 pathological M0 (pM0) carcinomas, 5 pM1a carcinomas, 2 pM1b carcinomas and 1 M1c carcinoma. The 5 pM1a carcinomas were due to pleural dissemination or malignant effusion at operation. The 2 pM1b carcinomas were due to single brain metastasis. The 1 M1c carcinoma was due to metastasis to the liver and stomach. Brain metastasis was diagnosed by brain enhanced MRI, and metastasis to liver and stomach was diagnosed by biopsy after FDG-PET/CT. There were 118 pStage IA carcinomas, 58 pStage IB carcinomas, 26 pStage IIA carcinomas, 21 pStage IIB carcinomas, 30 pStage IIIA carcinomas, 1 pStage IIIB carcinoma, 7 pStage IVA carcinomas and 1 pStage IVB carcinoma.

For 52 BPNMs, there were 41 inflammatory BPNMs [Mycobacterial disease 13 (tuberculosis 5, nontuberculous mycobacteria 8), pneumonia 13, pulmonary abscess 8, pulmonary scar 3, organized pneumonia 2, pulmonary granuloma 1 and sarcoidosis 1, and 11 non-inflammatory BPNMs (hamartoma 5, pulmonary sequestration 2, nodular lymphoid hyperplasia 1, inflammatory myofibroblastic tumor 1, encapsulated pleural effusion 1 and pleural cyst 1). Twenty-six BPNMs were diagnosed pathologically by resection. Thirteen BPNMs were diagnosed as mycobacterial disease by bacterial culture (3) or resection (10). The remaining 13 BPNMs were diagnosed as pneumonia by decreased size or disappearance of the BPNMs.

### 2.3. MR Imaging

All MR images were produced with a 1.5 T superconducting magnetic scanner (Magnetom Avanto, Siemens, Erlangen, Germany) with two anterior six-channel body phased-array coils and two posterior spinal clusters (six-channels each). The conventional MR images consisted of a coronal T1-weighted spin-echo sequence and coronal and axial T2-weighted fast spin-echo.

Examination of the 1.5-T MRI was performed as follows: T2WI was obtained in a turbo spin echo (TSE); TR/TE, 4400–6000/74 ms; FOV, 350 × 240 mm; matrix, 320 × 198; thickness, 6.0 mm), Flip angle 90°. T1-weighted imaging (T1WI) was obtained in gradient recalled echo (GRE) VIBE; TR/TE, 6.54/4.78 ms; FOV, 380 × 240 mm; matrix, 256 × 151; thickness, 3.5 mm). DWIs using a single-shot echo-planar method were applied with a slice thickness of 6 mm under SPAIR (spectral attenuated inversion recovery) with a respiratory triggered scan with the following parameter: TR/TE/flip angle, 3000–4500/65/90; diffusion gradient encoding in three orthogonal directions; *b* value = 0 and 800 s/mm^2^; field of view, 350 mm; matrix size, 128 × 128.

After image reconstruction, a two-dimensional (2D) round or elliptical region of interest (ROI) was drawn on the lesion which was detected visually on the ADC map with reference to T2-weighted or CT image. The procedures were repeated three times and the minimum ADC value was obtained. T2 contrast ratio (T2 CR) of a PNM was defined based on the definition of Koyama et al. [16]: T2 CR = the ratio of T2 signal intensity of a PNM divided by T2 signal intensity of a rhomboid muscle. The ROI placed over the muscle was fixed at 120 mm^2^ (a round of 8 mm in size) according to the description of Koyama et al. T2 signal intensities of PNMs were obtained by drawing round, elliptical or free-hand ROIs on lesions which were detected visually on the T2WI. The radiologist (M.D.) with 25 years of MRI experience who was unaware of the patients’ clinical data and one pulmonologist (K.U.) with 28 years of experience evaluated the MRI data. All measures were performed by one experienced author (K.U.) supported by the experienced radiologist (M.D.). They eventually reached the same consensus. There was no discrepancy in the data between the radiologist and the pulmonologist.

### 2.4. Statistical Analysis

The data are expressed as the mean ± standard deviation. A two-tailed Student t-test was performed for comparison of several values of two groups and ANOVA was performed for comparison of several values of three or more groups in several factors. A Chi-square test was used for the comparison of ratios. A receiver operating characteristics (ROC) curve was constructed to evaluate the diagnostic capability of the ADC value and T2 CR value in terms of malignant–benign differentiation. Using GraphPad Prism (Version 5.02, GraphPad Software, Inc. La Jolla, CA, USA) optimal cutoff values (OCVs) of ADC and T2 CR for a differential diagnosis were determined. The statistical analyses were performed using the computer software program StatView for Windows (Version 5.0; SAS Institute Inc. Cary, NC, USA). A *p*-value of <0.05 was considered statistically significant.

## 3. Results

In the ROC curve of ADC for all the PNMs, when the optimal cut off value (OCV) of ADC was set at 1.470 × 10^−3^ mm^2^/s, the area under the ROC curve was 74.8%, the sensitivity was 84.2%, and the specificity was 63.5% (Figure 1). In the ROC curve of T2 CR for all the PNMs, when the OCV of T2 CR was set at 2.45, the area under the ROC curve was 74.3%, the sensitivity was 89.5%, and the specificity was 65.4% (Figure 2).

In a Chest CT, DWI, ADC map and T2WI an adenocarcinoma and a squamous cell carcinoma are presented (Figure 3). The adenocarcinoma had 1.39 × 10^−3^ mm^2^/s of ADC (True Positive: TP) and 1.67 of T2 Contrast ratio (CR) (TP). The squamous cell carcinoma had 1.04 × 10^−3^ mm^2^/s of ADC (TP) and 1.25 of T2 CR (TP). In a Chest CT, DWI, ADC map and T2WI a hamartoma, a pulmonary abscess, and pulmonary tuberculosis are presented (Figure 4). The hamartoma had 2.43 × 10^−3^ mm^2^/s of ADC (True Negative: TN) and 3.61 of T2 CR (TN). The pulmonary abscess had 0.837 × 10^−3^ mm^2^/s of ADC (False Positive: FP) and 3.64 of T2 CR (TN). Pulmonary tuberculosis 1.85 × 10^−3^ mm^2^/s of ADC (TN) and had 1.87 of T2 CR (FP).

The ADC and T2 CR of lung cancers and BPNMs are shown in Figure 5. The ADC (1.24 ± 0.29 × 10^−3^ mm^2^/s) of lung cancers was significantly lower than that (1.69 ± 0.58 × 10^−3^ mm^2^/s) of BPNMs (*p* < 0.0001). The T2 CR (2.01 ± 0.52) of lung cancers was significantly lower than that (2.74 ± 1.02) of BPNMs (*p* < 0.0001).

ADC and T2 CR based on each diagnosis of each cell type of lung cancer and BPNM are shown in Figure 6. Although a pulmonary abscess is benign, its ADC value showed a lower ADC value (Figure 7). The ADC (1.20 ± 0.53 × 10^−3^ mm^2^/s) of pulmonary abscesses was not significantly lower than that (1.24 ± 0.29 × 10^−3^ mm^2^/s) of lung cancers (*p* = 0.695). The ADC (1.58 ± 0.47 × 10^−3^ mm^2^/s) of mycobacterial infections was significantly higher than that (1.24 ± 0.29 × 10^−3^ mm^2^/s) of lung cancers (*p* < 0.0001). On the other hand, the T2 CR (2.93 ± 1.26) of pulmonary abscesses was significantly higher than that (2.01 ± 0.52) of lung cancers (*p* = 0.010) and the T2 CR (2.41 ± 0.86) of mycobacterial infections was significantly higher than (2.01 ± 0.52) of lung cancers (*p* = 0.010).

Concerning the pathologic subtypes (Figure 8), the ADC (1.89 ± 0.36 × 10^−3^ mm^2^/s) of mucinous adenocarcinomas was significantly higher than that (1.27 ± 0.26 × 10^−3^ mm^2^/s) of acinar adenocarcinomas (*p* < 0.0001), that (1.27 ± 0.23 × 10^−3^ mm^2^/s) of papillary adenocarcinomas (*p* < 0.0001), than (1.16 ± 0.22 × 10^−3^ mm^2^/s) of lepidic adenocarcinomas (*p* < 0.0001), than (1.16 ± 0.14 × 10^−3^ mm^2^/s) of micropapillary adenocarcinomas (*p* < 0.0001), than (1.10 ± 0.20 × 10^−3^ mm^2^/s) of solid adenocarcinomas (*p* < 0.0001), than (1.16 ± 0.20 × 10^−3^ mm^2^/s) of squamous cell carcinomas (*p* < 0.0001), than (0.90 ± 0.16 × 10^−3^ mm^2^/s) of small cell carcinomas (*p* < 0.0001), than (1.21 ± 0.15 × 10^−3^ mm^2^/s) of adenosquamous carcinomas (*p* < 0.007), and than (0.95 ± 0.18 × 10^−3^ mm^2^/s) of carcinoids (*p* < 0.0034). 

Concerning T2WI, the T2CR (2.90 ± 0.83) of mucinous adenocarcinomas was significantly higher than that (1.92 ± 0.39) of acinar adenocarcinomas (*p* < 0.0001), that (1.92 ± 0.42) of papillary adenocarcinomas (*p* < 0.0001), than (1.52 ± 0.49) of lepidic adenocarcinomas (*p* < 0.0001), than (2.04 ± 0.32) of micropapillary adenocarcinomas (*p* = 0.017), than (2.01 ± 0.31) of solid adenocarcinomas (*p* = 0.0006), than (2.15 ± 0.40) of squamous cell carcinomas (*p* < 0.0001), and than (1.97 ± 0.16 × 10^−3^ mm^2^/s) of small cell carcinomas (*p* = 0.015).

Concerning diagnostic efficacy of DWI for the 324 PNMs, when the OCV (1.470 × 10^−3^ mm^2^/s) of ADC was applied for differential diagnosis, its sensitivity was 83.9% (220/262), its specificity was 63.4% (33/52), and its accuracy was 80.6% (253/314). Concerning diagnostic efficacy of T2WI for the 324 PNMs, when the OCV (2.45) of T2 CR was applied for differential diagnosis, its sensitivity was 89.7% (235/262), its specificity was 61.5% (32/52), and its accuracy was 85.0% (267/314). 

Diagnostic efficacy by both DWI and T2WI is presented in Table 2. In the 212 PNMs which were judged to be malignant by both DWI and T2WI, 203 PNMs (95.8%) were lung cancers. In 33 PNMs which were judged to be benign by both DWI and T2WI, 23 PNMs (69.7%) were BPNMs. When DWI and T2WI had supporting results, the sensitivity [95.8% (203/212)] of lung cancers by both DWI and T2WI was significantly higher than that [30.3% (10/33)] of lung cancers which were judged as benign by both DWI and T2WI (*p* < 0.001). The specificity [69.7% (23/33)] of BPNMs which were judged as benign by both DWI and T2WI was significantly higher than that [4.2% (9/212)] of BPNMs which were judged as malignant by both DWI and T2WI (*p* < 0.001). On the other hand, the remaining 69 PNMs had contradicting results with DWI and T2WI. When there were contradicting results, DWI was correct in 40.6% of patients (28/69), and T2WI was correct in 59.4% of patients (41/69).

## 4. Discussion

DWI utilizes the free diffusion of water molecules and recent developments in molecular imaging based on MR techniques have provided researchers and clinicians with new tools to improve most facets of cancer care [17]. Molecular imaging is described as imaging techniques used to detect molecular signatures at the cellular and gene expression levels [17]. MRI is a particularly attractive method for molecular imaging applications, due to its noninvasive nature, outstanding signal-to-noise ratio, high spatial resolution, exceptional tissue contrast, and short imaging times [18]. Koyama et al. [16] described that non-contrast-enhanced pulmonary MRIs can effectively detect malignant nodules as well as a thin-section multidetector CT (MDCT). MRI can detect and stage lung cancer, and this method could be an excellent alternative to CT or PET/CT in the investigation of pulmonary malignancies and other diseases [19]. Conventional MRI can reveal the essential differences between mass-like tuberculosis and lung cancer and may be helpful for discriminating pulmonary masses [20]. When an invasion is unclear by CT criteria, MRI can play an important role in defining lesser degrees of invasion [21]. MRI is superior to CT for the visualization of the pericardium, the heart and mediastinal vessels [22]. MRI can be of use specifically for assessing invasion of the superior vena cava or myocardium, or extension of the tumor into the left atrium via pulmonary veins [22]. Although PET-CT is believed to be more accurate for this purpose, MRI has the advantage of being more universally available and less expensive [19].

DWI has been described to be able to differentiate malignancy from benignity for PNMs [7,8,9]. DWI is more useful for the differentiation of SCLC from NSCLC than STIR [23]. The ADC value of adenocarcinoma was significantly higher than that of either squamous cell carcinoma or small cell carcinoma, which shows that the tissue cellularity of squamous cell carcinoma or small cell carcinoma would be higher than that of adenocarcinoma [14]. ADC histogram analysis can provide important information on tumor biology in cervical cancer [24]. ADC histograms which analyze the whole tumor was reported to be useful for malignancy evaluation [25,26]. The pulmonary abscess has strongly restricted diffusion in DWI. Some pathological processes such as sarcoidosis, lung abscess, chronic pneumonia, pulmonary tuberculosis, nontuberculous mycobacteria, scars, and other inflammatory or infectious conditions behave like malignant lesions by exhibiting diffusion restriction [27,28,29]. On the other hand, carcinomas have restricted diffusion because of their high cellular proliferation rate, cells with a large size nucleus, intracellular macromolecules, high nucleus–cytoplasm rate and limited size of the extracellular matrix [30,31].

Particularly the T2WI sequence may help discriminate between progressive massive fibrosis (PMF) and lung cancer [32]. T2WI expresses liquid in the lesion brightly, and is suitable for detection of pulmonary abscess, chronic pneumonia, pulmonary tuberculosis, nontuberculous mycobacteria, and other inflammatory or infectious conditions. In this study, the T2 CR (2.93 ± 1.26) of pulmonary abscesses was significantly higher than that (2.01 ± 0.52) of lung cancers (*p* = 0.010), which proved that T2WI was useful for discriminating pulmonary abscess from lung cancer. The signal intensity ratios (SIRs) of the lesion divided by the rhomboid muscle on T2WI and T1WI were significantly different between mass-like tuberculosis and lung cancer [20]. Traditional T2WI can detect effusion in the body or in the tumor as a high signal intensity (showing white). Pleural effusion and cystic mediastinal tumors are easy to diagnose with T2WI.

We focused on not only the strengths of DWI but also the strengths of T2WI. The absolute most accurate way to reduce false positives and false negatives is to perform a biopsy. However, by using DWI and T2WI before the minimally invasive options such as serology (autoantibodies), gene chip expression platforms, and next-generation optical genome editing we will be able to get a more accurate diagnosis compared to the previous methods of CT or FDG-PET/CT. The deployment of non-invasive cancer diagnostic tools such as the use of DWI in concert with T2WI can offer potential advantages for both patients and providers. Although the diagnostic capability of both DWI plus T2WI is increasingly being used for PNMs one must take into account the several cases of false positives as well as false negatives by both diagnostic tools.

In this study, the ROC curve showed the diagnostic performance of DWI for distinguishing BPNM from lung cancer. The area under the ROC curve 74.8% and the sensitivity was 84.2%, and the specificity was 63.5% when the OCV of ADC was at 1.470 × 10^−3^ mm^2^/s. The diagnostic efficacy was not a high value although the ADC (1.24 ± 0.29 × 10^−3^ mm^2^/s) of lung cancer was significantly lower than that (1.69 ± 0.58 × 10^−3^ mm^2^/s) of BPNMs. The ROC curve showed the diagnostic performance of T2 CR for distinguishing BPNM from lung cancer. The area under the ROC curve 74.3% and the sensitivity was 89.5%, and the specificity 65.4% when the OCV of T2 CR was at 2.45. The diagnostic efficacy was not a high value although the T2 CR (2.01 ± 0.52) of lung cancer was significantly lower than that (2.74 ± 1.02) of BPNMs (*p* < 0.0001). Combination assessment of DWI and T2WI, in 212 PNMs which were judged to be malignant by both DWI and T2WI, 203 PNMs (95.8%) were lung cancers. In 33 PNMs which were judged to be benign by both DWI and T2WI, 23 PNMs (69.7%) were BPNMs. The remaining 69 PNMs had contradicting results with DWI and T2WI. When there were contradicting results, DWI was correct in 40.6% of patients (28/69), and T2WI was correct in 59.4% of patients (41/69). In cases where DWI and T2WI give contradicting results, additional examinations should be given to clarify. The combined assessment of DWI and T2WI could judge PNMs more precisely and would be acceptable for differential diagnosis of PNMs. In the literature, there were several articles concerning the diagnostic performance of DWI and T2WI for differential diagnosis for many other organs of the body, but there is no published research that focused on the lungs. This is the first paper that dealt with the combined diagnostic ability of DWI plus T2WI for differential diagnosis of PNMs. Diagnostic possibilities would be increased after fused T2WI and DWI for lung cancers and BPNMs. T2WI combined with DWI may be a valuable tool for detecting prostate cancer in the overall evaluation of prostate cancer [33,34], and myometrial invasion and staging of endometrial carcinoma [35,36]. Adding DWI to T2WI is helpful for detecting viable tumors after neoadjuvant chemoradiation therapy compared with T2WI alone or PET/CT in patients with locally advanced rectal cancer [37]. Furthermore, combining T2WI, DWI and ADC values provides increased accuracy for differentiation between benign and malignant lesions, compared with DCE (dynamic contrast-enhanced)-MRI [38].

MRI involves no radiation exposure as well as no contrast mediums and is suitable and ideal for the examination of pregnant women and children. In the next decade, MRI will be available more for PNM assessment because CT or FDG-PET/CT has some risk of radiation exposure which has to be explained to patients, who often find this unexpected. Mark L. Scheibel [39] mentioned that DWI can be used to adequately stage non-small cell lung cancer (NSCLC). If whole-body DWI can be shown to have equipoise with FDG-PET/CT for the clinical staging of NSCLC, this would reduce the costs of patient workup because 18F-FDG PET would no longer be needed. He mentioned that, in the near future, only whole-body DWI will be needed for clinical staging in patients with a new diagnosis of NSCLC.

We should keep in mind that the study had two limitations. First, it was conducted at a single institution. Further, adequately powered prospective randomized trials will be necessary to evaluate the combined assessment of DWI and T2WI for differentiating between lung cancer and BPNM.

## 5. Conclusions

In the 212 PNMs which were judged to be malignant by both DWI and T2WI, 203 PNMs (95.8%) were lung cancers. In the 33 PNMs which were judged to be benign by both DWI and T2WI, 23 PNMs (69.7%) were BPNMs. The combined assessment of DWI and T2WI could judge PNMs more precisely and would be acceptable for differential diagnosis of PNMs.

## Figures and Tables

**Figure 1 cancers-13-01551-f001:**
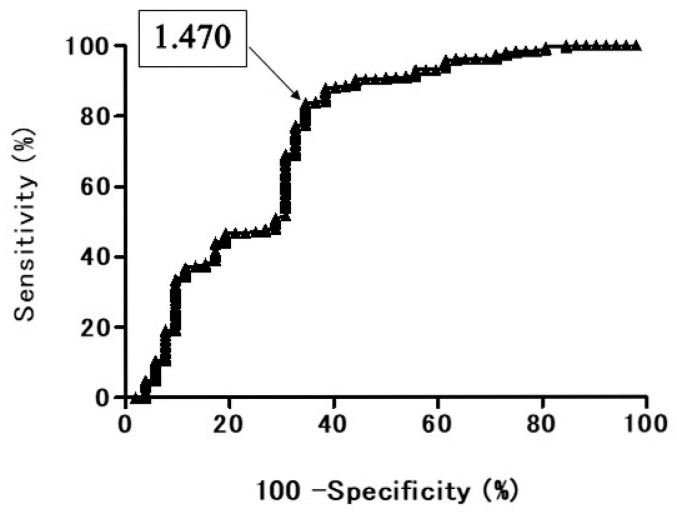
Receiver operating characteristic (ROC) curve shows the diagnostic performance of diffusion-weighted magnetic resonance imaging (DWI) for distinguishing benign pulmonary nodule and mass (BPNM) from lung cancer. Area under the ROC curve 74.8%. Apparent diffusion coefficient (ADC) = 1.470 × 10^−3^ mm^2^/s, sensitivity 84.2%, specificity 63.5%.

**Figure 2 cancers-13-01551-f002:**
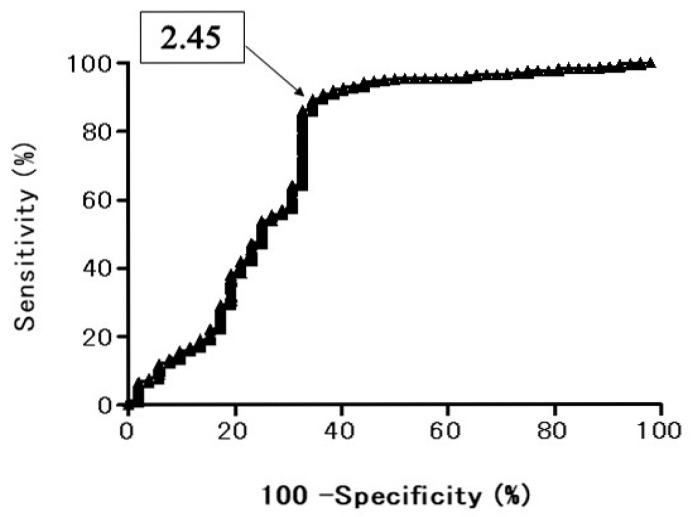
Receiver operating characteristic (ROC) curve shows the diagnostic performance of T2 contrast ratio (T2 CR) for distinguishing BPNM from lung cancer. Area under the ROC curve 74.3%. T2 CR = 2.45, sensitivity 89.5%, specificity 65.4%.

**Figure 3 cancers-13-01551-f003:**
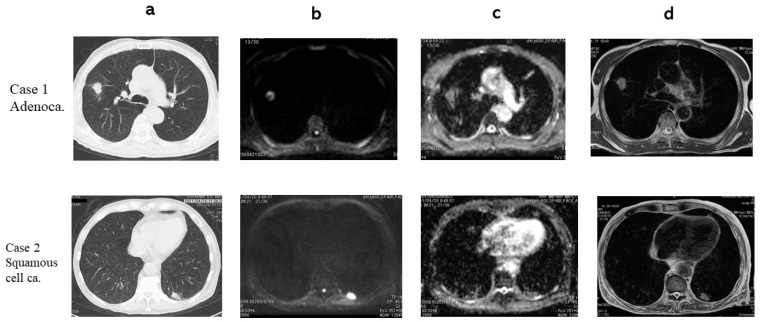
(**a**) CT, (**b**) DWI, (**c**) ADC map, (**d**) T2 WI. Case 1: Adenocarcinoma. ADC 1.39 × 10^−3^ mm^2^/s, T2 CR: 1.67. Case2: Squamous cell carcinoma. ADC 1.04 × 10^−3^ mm^2^/s, T2 CR: 1.25.

**Figure 4 cancers-13-01551-f004:**
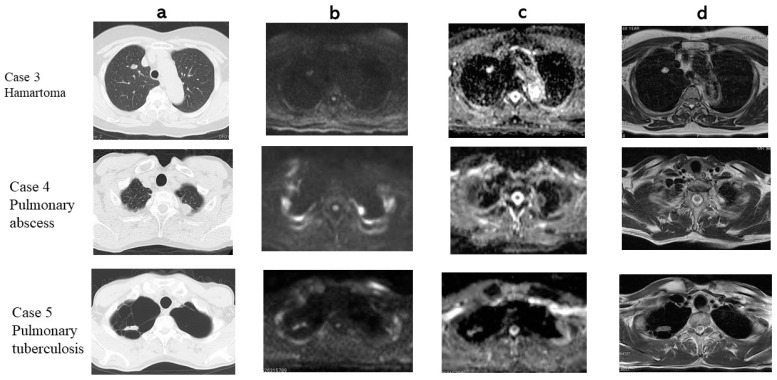
(**a**) CT, (**b**) DWI, (**c**) ADC map, (**d**) T2 WI. Case 3: Hamartoma, ADC 2.43 × 10^−3^ mm^2^/s, T2 CR: 3.61. Case 4: Pulmonary abscess. ADC 0.837 × 10^−3^ mm^2^/s, T2 CR: 3.64. Case 5: Pulmonary tuberculosis. ADC 1.85 × 10^−3^ mm^2^/s, T2 CR:1.87.

**Figure 5 cancers-13-01551-f005:**
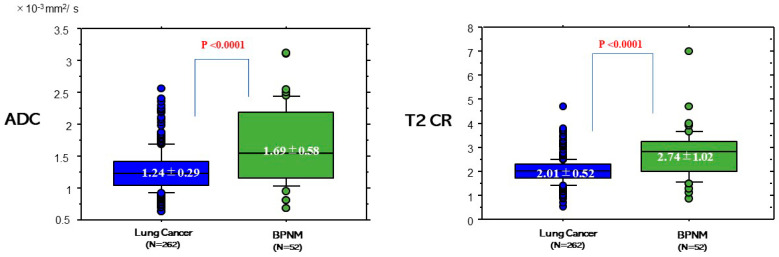
ADC and T2 Contrast ratio (CR) between lung cancers and BPNMs. The ADC (1.24 ± 0.29 × 10^−3^ mm^2^/s) of lung cancers was significantly lower than that (1.69 ± 0.58 × 10^−3^ mm^2^/s) of BPNMs (*p* < 0.0001). T2 CR was the ratio of T2 signal intensity of the pulmonary nodule divided by T2 signal intensity of a rhomboid muscle. The T2 CR (2.01 ± 1.02) of lung cancers was significantly lower than that (2.74 ± 1.02) of BPNMs (*p* < 0.0001).

**Figure 6 cancers-13-01551-f006:**
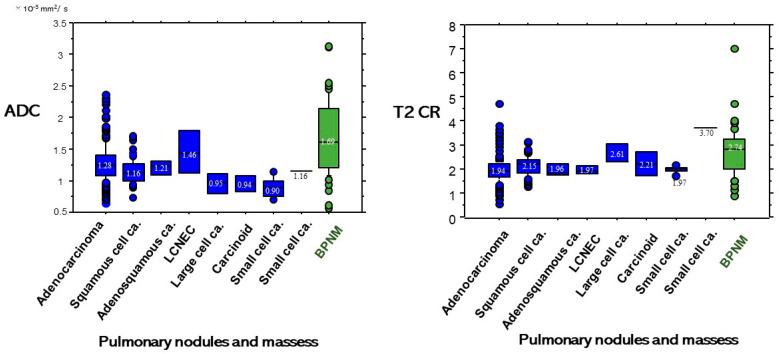
ADC and T2 CR based on diagnosis of PNMs.

**Figure 7 cancers-13-01551-f007:**
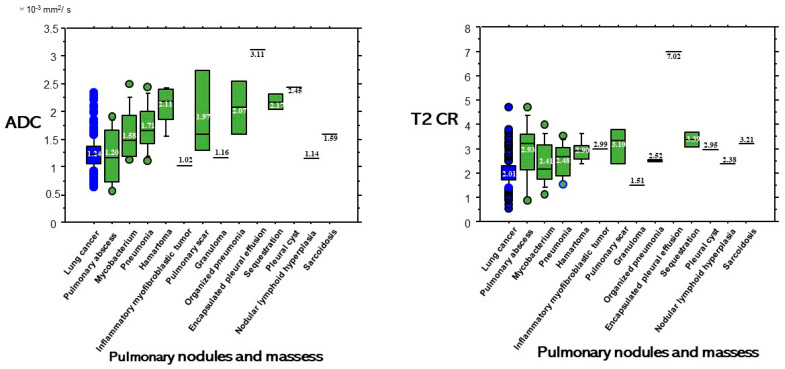
The ADC (1.20 ± 0.53 × 10^−3^ mm^2^/s) of pulmonary abscesses was not significantly lower than that (1.24 ± 0.29 × 10^−3^ mm^2^/s) of lung cancers (*p* = 0.695). The ADC (1.58 ± 0.47 × 10^−3^ mm^2^/s) of mycobacterial infections was significantly higher than that (1.24 ± 0.29 × 10^−3^ mm^2^/s) of lung cancers (*p* < 0.0001). The T2 CR (2.93 ± 1.26) of pulmonary abscesses was significantly higher than that (2.01 ± 0.52) of lung cancers (*p* = 0.010) and the T2 CR (2.41 ± 0.86) of mycobacterial infections was significantly higher than (2.01 ± 0.52) of lung cancers (*p* = 0.010).

**Figure 8 cancers-13-01551-f008:**
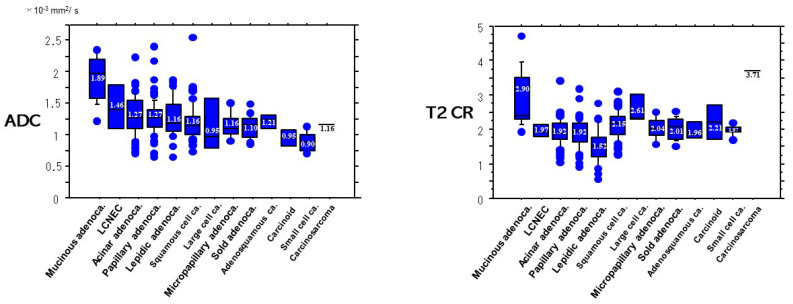
ADC and T2 CR based on pathologic subtypes of lung cancer. Both the ADC and the T2CR of mucinous adenocarcinoma were significantly higher than those of other pathologic subtypes.

**Table 1 cancers-13-01551-t001:** Patients’ clinical data.

	Lung Cancer		Benign Pulmonary Nodule and Mass (BPNM)		
No. of patients		262			52
Male/Female		164/98			32/20
Cell type	adenocarcinoma	183			
	squamous cell ca.	60			
	LCNEC	4			
	Large cell ca.	3			
	Adenosquamous ca.	3			
	Carcinoid	2			
	Small cell ca.	6			
	Carcinosarcoma	1			
pT	T1mi	2			
	T1a	70			
	T1b	49			
	T1c	3			
	T2a	78			
	T2b	19			
	T3	35			
	T4	6			
pN	N0	209			
	N1	33			
	N2	20			
pM	M0	254			
	M1a	5			
	M1b	2			
	M1c	1			
pStage	Stage IA	118			
	Stage IB	58			
	Stage IIA	26			
	Stage IIB	21			
	Stage IIIA	30			
	Stage IIIB	1			
	Stage IVA	7			
	Stage IVB	1			
Causes of BPNM			Inflammatory	Mycobacterial disease	13
				Pneumonia	13
				Pulmonary abscess	8
				Pulmonary scar	3
				Organized pneumonia	2
				Other	2
			Non-inflammatory	Hamartoma	5
				Pulmonary sequestration	2
				Other	4
Diagnosis and therapy	Resection	262		Resection	36
				Bacterial culture	3
				Decreased size or disappearance	13

**Table 2 cancers-13-01551-t002:** Diagnostic efficacy by DWI and T2WI.

		Diagnosis	No. of Cases	Supporting Results		Contradicting Results	
				DWI: Malignant T2WI: Malignant	DWI: Benign T2WI: Benign	DWI: Malignant T2WI: Benign	DWI: Benign T2WI: Malignant
Diagnosis	Malignant	Mucinous AD	13	1 (7.7%)	6	0	6
		Acinar AD	58	44 (75.9%)	2	0	12
		Papillary AD	62	53 (85.5%)	0	2	7
		Lepidic AD	27	25 (92.6%)	1	0	1
		Micropapillary AD	7	6 (85.7%)	0	1	0
		Solid AD	16	14 (87.5%)	0	1	1
		Adenosquamous ca.	3	3 (100%)	0	0	0
		Squamous cell ca.	60	46 (76.7%)	1	10	3
		LCNEC	4	2 (50%)	0	0	2
		Large cell ca.	3	2 (66.7%)	0	1	0
		Carcinoid	2	1 (50%)	0	1	0
		Small cell ca.	6	6 (100%)	0	0	0
		Carcinosarcoma	1	0 (0%)	0	1	0
		Total	262	203 (95.8%)	10 (30.3%)	17 (65.4%)	32 (74.4%)
		Pneumonia/ Organized pneumonia	15	2	7 (46.7%)	2	4
		Mycobacterial infection	13	2	2 (15.4%)	4	5
		Pulmonary abscess	8	2	3 (37.5%)	2	1
	Benign	Pulmonary scar/ Puimonary granuloma	4	2	2 (50%)	0	0
		Sarcoidosis	1	0	1 (100%)	0	0
		Hamartoma	5	0	5 (100%)	0	0
		Pulmonary sequestration	2	0	1 (50%)	0	1
		Other diseases	4	1	2 (505)	1	0
		Total	52	9 (4.2%)	23 (69.7%)	9 (34.6%)	11 (25.6%)
Total			314	212 (100%)	33 (100%)	26 (100%)	43 (100%)

## Data Availability

The data presented in this study are available on request from the corresponding author.

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
