# Peer review of "Combination Assessment of Diffusion-Weighted Imaging and T2-Weighted Imaging Is Acceptable for the Differential Diagnosis of Lung Cancer from Benign Pulmonary Nodules and Masses"

_cancers, 2021, doi:10.3390/cancers13071551_

Round 1
Reviewer 1 Report
The Reviewer finds the submitted Manuscript to be of generally high quality. Certainly, the deployment of non-invasive cancer diagnostic tools, such as the use of DWI in concert with T2WI, offer potential advantages for both patients and providers. Here however, the Reviewer suggests that the Authors might more clearly articulate a key point. That is, DWI plus T2WI [even used in conjunction with one another] appear to result in a significant percentage of false positives as well as false negatives.
Moreover, though biopsy often represents the gold standard in histological detection of neoplastic pathology, aside from DWI there are presently a number of minimally invasive options that may produce meaningful diagnostic output. These include serology (autoantibodies), gene chip expression platforms and most recently, next-generation optical genome mapping. Given the core message of selecting for non-invasive analyses, it is the Reviewer's viewpoint that the aforementioned methods ought at least be mentioned.
Author Response
Reviewer 1
Comments and Suggestions for Authors
The Reviewer finds the submitted Manuscript to be of generally high quality. Certainly, the deployment of non-invasive cancer diagnostic tools, such as the use of DWI in concert with T2WI, offer potential advantages for both patients and providers. Here however, the Reviewer suggests that the Authors might more clearly articulate a key point. That is, DWI plus T2WI [even used in conjunction with one another] appear to result in a significant percentage of false positives as well as false negatives.
I appreciate the reviewer’s comments and advice. I added these points in the discussion: The deployment of non-invasive cancer diagnostic tools such as the use of DWI in concert with T2WI can offer potential advantages for both patients and providers. Although the diagnostic capability of both DWI plus T2WI are increasingly being used for PNMs one must take into account the several cases of false positives as well as false negatives by both diagnostic tools
Moreover, though biopsy often represents the gold standard in histological detection of neoplastic pathology, aside from DWI there are presently a number of minimally invasive options that may produce meaningful diagnostic output. These include serology (autoantibodies), gene chip expression platforms and most recently, next-generation optical genome mapping. Given the core message of selecting for non-invasive analyses, it is the Reviewer's viewpoint that the aforementioned methods ought at least be mentioned.
The absolute most accurate way to reduce false positives and false negatives is to preform a biopsy. However, by using DWI and T2WI before the minimally invasive options such as serology (autoantibodies), gene chip expression platforms, and next-generation optical genome editing we will be able to get a more accurate diagnosis compared to the previous methods of CT or FDG-PET/CT.

Reviewer 2 Report
Thank you for the opportunity to review this interesting paper.
The authors performed a large imaging analysis to discriminate benign from malignant pulmonary lesions based upon 314 lesions. The addition to T2-weighted ratio helped to discriminate abscesses from malignant lesions, which is a known limitation of DWI alone.
The results are interesting and timely.
The manuscript needs an Editing of the language. Some passages are hard to understand.
Please add into the materials and methods part, whether these patients were also used in other papers of your work group and cite the papers (for example10.3390/cancers12051194 )
Could you provide an interreader agreement for a subset of your patients? This might differ between the ADC values and T2-ratios.
You could add other references regarding the topic of histopathology features and ADC values.
Author Response
Reviewer 2
Comments and Suggestions for Authors
Thank you for the opportunity to review this interesting paper.
The authors performed a large imaging analysis to discriminate benign from malignant pulmonary lesions based upon 314 lesions. The addition to T2-weighted ratio helped to discriminate abscesses from malignant lesions, which is a known limitation of DWI alone.
The results are interesting and timely.
The manuscript needs an Editing of the language. Some passages are hard to understand.
Mr. Dustin Keeling, whose native language is English, has proofread this paper.
Please add into the materials and methods part, whether these patients were also used in other papers of your work group and cite the papers (for example10.3390/cancers12051194 )
Out of 262 patients with lung cancer 226 of those patients that were enrolled in this study were also used in another paper ( 14. Usuda K, Iwai S, Yamagata A, Seizure A, Motono N, Matoba M, Doai M, Yamada S, Ueda Y, Hirata K, Uramoto H. Relationships and Qualitative Evaluation Between Diffusion-Weighted Imaging and Pathologic Findings of Resected Lung Cancers.Cancers (Basel). 2020;12(5):1194. doi:10.3390/cancers12051194).
Could you provide an interreader agreement for a subset of your patients? This might differ between the ADC values and T2-ratios.
The experienced radiologist (M.D.) with 25 years of MRI experience who was unaware of the patients’ clinical data and one pulmonologist (K.U.) with 28 years of experience evaluated the MRI data. All measures were performed by one experienced author (K.U.) supported by the experienced radiologist (M.D.). They eventually reached the same consensus. There was no discrepancy in the data between the radiologist and the pulmonologist.
You could add other references regarding the topic of histopathology features and ADC values.
I added the next references.
DWI is more useful for the differentiation of SCLC from NSCLC than STIR. (23. Koyama, H.; Ohno, Y.; Nishio, M.; Takenaka, D.; Yoshikawa, T.; Matsumoto, S.; Seki,S.; Maniwa,Y.; Ito, T.; Nishimura, Y.; Sugimura, K. Diffusion-weighted imaging vs STIR turbo SE imaging: capability for quantitative differentiation of small-cell lung cancer from non-small-cell lung cancer. Br J Radiol. 2014; 87(1038): 20130307. doi: 10.1259/bjr.20130307
The ADC value of adenocarcinoma was significantly higher than that of either squamous cell carcinoma or small cell carcinoma, which shows that the tissue cellularity of squamous cell carcinoma or small cell carcinoma would be higher than that of adenocarcinoma (14. Usuda K, Iwai S, Yamagata A, Sekimura A, Motono N, Matoba M, Doai M, Yamada S, Ueda Y, Hirata K, Uramoto H.Relationships and Qualitative Evaluation Between Diffusion-Weighted Imaging and Pathologic Findings of Resected Lung Cancers.Cancers (Basel). 2020;12(5):1194. doi:10.3390/cancers12051194).
ADC histogram analysis can provide important information on tumor biology in cervical cancer (24. Schob, S.; Meyer, H.J.; Pazaitis, N.; Schramm, D.; Bremicker, K.; Exner, M.; Höhn, A.K.; Garnov, N.; Surov, A.; Schob, S.; et al. ADC Histogram Analysis of Cervical Cancer Aids Detecting Lymphatic Metastases-a Preliminary Study. Mol. Imaging Biol. 2017, 19, 953-962. doi: 10.1007/s11307-017-1073-y.)
ADC histograms which analyze the whole tumor was reported to be useful for malignancy evaluation. ( 25. Xue, H.; Ren, C.; Yang, J.; Sun, Z.; Li, S.; Jin, Z.; Shen, K.; Zhou, W. Histogram Analysis of Apparent Diffusion Coefficient for the Assessment of Local Aggressiveness of Cervical Cancer. Arch. Gynecol. Obstet. 2014, 290, 341-348. doi: 10.1007/s00404-014-3221-9. 26. Donati, O.F.; Mazaheri, Y.; Faq , A.; Vargas, H.A.; Zheng, J.; Moskowitz, C.S.; Hricak, H.; Akin, O. Prostate Cancer Aggressiveness: Assessment With Whole-Lesion Histogram Analysis of the Apparent Diffusion Coefficient. Radiology 2014, 271, 143-152. doi: 10.1148/radiol.13130973.)

Reviewer 3 Report
The diagnosis of lung cancer, especially of small diameter and advancement, is problematic. Therefore, a discussion on the subject of non-invasive tests that may positively improve this situation is very much needed. The work is very valuable, but there are a few things that need to be clarified:
1. What examination was the primary diagnosis of lung tumors (x ray, CT?)
2. The authors write that some patients had metastases (M1a and M1b). How was it diagnosed, what was the metastasis site of M1b - where did the tumor spread to?
3. Clinical data (TNM, patient age, diagnosis, treatment) could be presented in a table.
4. The results in Table 1 could be statistically analyzed (chi square or Fisher's test).
Author Response
Reviewer 3
Comments and Suggestions for Authors
The diagnosis of lung cancer, especially of small diameter and advancement, is problematic. Therefore, a discussion on the subject of non-invasive tests that may positively improve this situation is very much needed. The work is very valuable, but there are a few things that need to be clarified:
1. What examination was the primary diagnosis of lung tumors (x ray, CT?)
Patients who had a lung cancer or a BPNM in chest X-ray were examined with plain CT first. The primary diagnosis of a lung cancer or a BPNM was performed by chest CT.
2. The authors write that some patients had metastases (M1a and M1b). How was it diagnosed, what was the metastasis site of M1b - where did the tumor spread to?
In the 262 patients that underwent surgery for lung cancer there were 5 pM1a carcinomas, 2 pM1b carcinomas and 1 M1c carcinoma. The 5 pM1a carcinomas was due to pleural dissemination or malignant effusion at operation. The 2 pM1b carcinomas was due to single brain metastasis. The 1 M1c carcinoma was due to metastasis to liver and stomach. Brain metastasis was diagnosed by brain enhanced MRI, and metastasis to liver and stomach was diagnosed by FDG-PET/CT.
- Clinical data (TNM, patient age, diagnosis, treatment) could be presented in a table.
In table 1 I included more precise data on the patients in the study (Sex, diagnosis, TNM, and treatment).
|
|
Table 1. Patients clinical data |
||||
|
  |
Lung cancer |
Benign pulmonary nodule and mass (BPNM) |
|||
|
No. of patients |
  |
262 |
  |
  |
52 |
|
Male / Female |
  |
164 / 98 |
  |
  |
32 / 20 |
|
Cell type |
adenocarcinoma |
183 |
|||
|
squamous cell ca. |
60 |
||||
|
LCNEC |
4 |
||||
|
Large cell ca. |
3 |
||||
|
Adenosquamous ca. |
3 |
||||
|
Carcinoid |
2 |
||||
|
Small cell ca. |
6 |
||||
|
  |
Carcinosarnoma |
1 |
  |
  |
  |
|
pT |
T1mi |
2 |
|||
|
T1a |
70 |
||||
|
T1b |
49 |
||||
|
T1c |
3 |
||||
|
T2a |
78 |
||||
|
T2b |
19 |
||||
|
T3 |
35 |
||||
|
  |
T4 |
6 |
  |
  |
  |
|
pN |
N0 |
209 |
|||
|
N1 |
33 |
||||
|
  |
N2 |
20 |
  |
  |
  |
|
pM |
M0 |
254 |
|||
|
M1a |
5 |
||||
|
M1b |
2 |
||||
|
  |
M1c |
1 |
  |
  |
  |
|
pStage |
Stage IA |
118 |
|||
|
Stage IB |
58 |
||||
|
Stage IIA |
26 |
||||
|
Stage IIB |
21 |
||||
|
Stage IIIA |
30 |
||||
|
Stage IIIB |
1 |
||||
|
Stage IVA |
7 |
||||
|
  |
Stage IVB |
1 |
  |
  |
  |
|
Causes of BPNM |
Inflammatory |
Mycobacterial disease |
13 |
||
|
Pneumonia |
13 |
||||
|
Pulmonary abscess |
8 |
||||
|
Pulmonary scar |
3 |
||||
|
Organized pneumonia |
2 |
||||
|
Other |
2 |
||||
|
Non-inflammatory |
Hamartoma |
5 |
|||
|
Pulmonary sequestration |
2 |
||||
|
  |
  |
  |
  |
Other |
4 |
|
Diagnosis and therapy |
Resection |
262 |
Resection |
36 |
|
|
Bacterial culture |
3 |
||||
|
  |
  |
  |
  |
Decreased size or disappearance |
13 |
The results in Table 1 could be statistically analyzed (chi square or Fisher's test).
I analyzed Table 1 with chi-square test.
In the 212 PNMs which were judged to be malignant by both DWI and T2WI, 203 PNMs (95.8%) were lung cancers. In 33 PNMs which were judged to be benign by both DWI and T2WI, 23 PNMs (69.7%) were BPNMs. When DWI and T2WI had supporting results, the sensitivity [95.8% (203/212)] of lung cancers by both DWI and T2WI was significantly higher than that [30.3% (10/33)] of lung cancers which were judged as benign by both DWI and T2WI (P<0.001). The specificity [69.7% (23/33)] of BPNMs which were judged as benign by both DWI and T2WI was significantly higher than that [4.2% (9/212)] of BPNMs which were judged as malignant by both DWI and T2WI (P<0.001). On the other hand, the remaining 69 PNMs had a contradicting results with DWI and T2WI. When there were contradicting results, DWI was correct in 40.6% of patients (28/69), and T2WI was correct in 59.4% of patients (41/69). In cases where DWI and T2WI give contradicting results, additional examinations should be given to clarify.
